# What We Know So Far about the Metabolite-Mediated Microbiota-Intestinal Immunity Dialogue and How to Hear the Sound of This Crosstalk

**DOI:** 10.3390/metabo11060406

**Published:** 2021-06-21

**Authors:** Clément Caffaratti, Caroline Plazy, Geoffroy Mery, Abdoul-Razak Tidjani, Federica Fiorini, Sarah Thiroux, Bertrand Toussaint, Dalil Hannani, Audrey Le Gouellec

**Affiliations:** 1Faculty of Medicine, CNRS, Grenoble INP, CHU Grenoble-Alpes, University Grenoble Alpes, TIMC (UMR5525), 38000 Grenoble, France; clement.caffaratti@univ-grenoble-alpes.fr (C.C.); cplazy@chu-grenoble.fr (C.P.); gmery@chu-grenoble.fr (G.M.); abdoul-razak.tidjani@univ-grenoble-alpes.fr (A.-R.T.); Sarah.Thiroux@univ-grenoble-alpes.fr (S.T.); btoussaint@chu-grenoble.fr (B.T.); 2Service de Biochimie Biologie Moléculaire Toxicologie Environnementale, UM Biochimie des Enzymes et des Protéines, Institut de Biologie et Pathologie, CHU Grenoble-Alpes, 38000 Grenoble, France; 3Plateforme de Métabolomique GEMELI-GExiM, Institut de Biologie et Pathologie, CHU Grenoble-Alpes, 38000 Grenoble, France; ffiorini@chu-grenoble.fr; 4Department of Infectiology-Pneumology, CHU Grenoble-Alpes, 38000 Grenoble, France

**Keywords:** microbiota, metabolites, immune system, host-microbiota crosstalk, non-targeted metabolomics

## Abstract

Trillions of microorganisms, termed the “microbiota”, reside in the mammalian gastrointestinal tract, and collectively participate in regulating the host phenotype. It is now clear that the gut microbiota, metabolites, and intestinal immune function are correlated, and that alterations of the complex and dynamic host-microbiota interactions can have deep consequences for host health. However, the mechanisms by which the immune system regulates the microbiota and by which the microbiota shapes host immunity are still not fully understood. This article discusses the contribution of metabolites in the crosstalk between gut microbiota and immune cells. The identification of key metabolites having a causal effect on immune responses and of the mechanisms involved can contribute to a deeper insight into host-microorganism relationships. This will allow a better understanding of the correlation between dysbiosis, microbial-based dysmetabolism, and pathogenesis, thus creating opportunities to develop microbiota-based therapeutics to improve human health. In particular, we systematically review the role of soluble and membrane-bound microbial metabolites in modulating host immunity in the gut, and of immune cells-derived metabolites affecting the microbiota, while discussing evidence of the bidirectional impact of this crosstalk. Furthermore, we discuss the potential strategies to hear the sound of such metabolite-mediated crosstalk.

## 1. Introduction

The human body is in a complex equilibrium with its microbial flora, and their synergistic interactions have been and still are the object of intense investigation. In 1991, Dr. Lynn Margulis introduced the concept of “holobiont” in her book” Symbiosis as a Source of Evolutionary Innovation”; according to this definition, the human body can be described as an assemblage of the host and the many species living in or around it [1]. The components of a holobiont are individual species or bionts, while the genomes of these bionts, including the human genome, the virome, the mycobiome, the microbiome, together define the hologenome. For decades, the development of genomic approaches, including phylogenetic marker-based microbiome profiling and shotgun metagenomics, has enabled a descriptive characterization of the microbiota composition and numerous links between its composition and diseases [2,3]. In adulthood, more than 1000 bacterial species have been identified, which shows the complexity of this large microbial community. Metagenomics gives access to the characterization of the microbiota at the taxonomic level, and at the level of the putative functions encoded by the numerous microbial genes. The recent advent of systems biology makes it possible to consider functional analysis (also simply called function) of the microbial community. This means, in particular, to quantify the metabolic activity, thanks to the measurement of RNA, by metatranscriptomics [4], of proteins, by metaproteomics [5], and finally, more recently, of metabolites, by metabolomics. Although some of the tools necessary for these integrative studies are still under development, they are essential to better understand the molecular mechanisms involved in symbiosis, but also related to the emergence of disease in the case of dysbiosis.

The human immune system plays an essential role in the development and education of innate and adaptive immunity after birth [6], and in maintaining homeostasis between humans and their resident microbial communities, thereby ensuring that the symbiotic nature of the host-microbe relationship is maintained. At the same time, commensal bacteria profoundly shape immunity [7,8,9]. The functions of the immune system could even include promoting the growth of beneficial microbes, as well as limiting the growth of harmful microbes, since the same microbe could be harmful or beneficial depending on the context (body sites, physiological state of the host, etc.). The crosstalk between immune cells and the microbiota engages a bidirectional communication, which has been studied for many years. Pattern Recognition receptors, especially Toll-like receptors (TLRs), initially described to recognize microbial signals during infection, are now known to be important players in shaping the gut microbiota and triggering a situation-specific immune response [9]. Recently, researchers have proposed that the interaction between gut microbiota and immunity is highly diverse and dynamic, and have demonstrated the cornerstone role that metabolites play in the evolutionary connection between symbiotic microbes and their hosts [10]. It is therefore important to study the content of gut metabolites and try to better characterize this dialogue (who are the players involved in the production of these metabolites, under what circumstances, and with which effects). According to Goodacre, “Metabolomics experiments aim to quantify all metabolites in a cellular system (cell or tissue) in defined states and at different times, so that the dynamics of any biotic, abiotic or genetic perturbation can be accurately assessed” and he positions metabolomics as an important systems biology approach to study the concepts of supraorganism or holobiont [11]. Despite the difficulty, metabolomics studies have clarified the metabolic profiles in the gut lumen and proved that the gut microbiome strongly influences the metabolome of the colonic lumen [12]. Microbiota-derived metabolites have been identified as components of the human metabolome [13]; some of these are recognized by immune cells via membrane or nuclear receptors and trigger a cascade of events leading to the maintenance of tolerance in the gut or the activation of antimicrobial strategies. These metabolites have a local effect in the gut on intestinal epithelial cells (IEC) or immune cells after diffusion, or are released systemically, impacting many host biological functions [13].

In a healthy state, the gut is a compartmentalized system that separates microbes from the host. Control of the microbiota biomass in the gut lumen relies on the integrity of the epithelial barrier and the effectiveness of the immune system in eliminating incoming pathogens penetrating the lamina propria. However, in this apparently hermetic system, there is constant communication between each of the players. A defective interaction between them is implicated in the pathogenesis, development, or severity of several diseases.

The present contribution attempts to provide a state-of-the-art description of the bidirectional interactions between the gut microbiota and the immune system and focuses on metabolites, defined as small organic molecules, intermediates, and products of an organism’s metabolism (<1.5 kDa). First, we review the most compelling microbiota-related metabolites associated with local or systemic immune modulations and the immune pathways that are triggered in the gut. Next, we present the traditional strategies used by the immune system to control the biomass and composition of the microbiota. Finally, we discuss the challenges and perspectives of metabolomics strategies and integrative tools in studying the crosstalk between immune cells and microbiota.

## 2. Microbiota-Derived Metabolites That Modulate Host Immunity in the Gut

The number of metabolites in the gut lumen is currently unknown and includes molecules of dietary, host, and microbial origin. Microbiota-derived metabolites that stimulate or suppress the immune response are either soluble bioactive molecules or membrane-bound compounds. It is already well-known that “immunometabolites”, such as succinate, itaconate, acetyl-CoA, and 2-hydroxyglutarate, to name a few, serve as signal transducers to regulate the immune cell function and disease outcome. Here, we discuss the role of the main gut immunometabolites currently known, and we provide a brief description of the classification, function, and mechanism by which they are involved in immune dysregulation (see the Appendix A).

### 2.1. Soluble Microbially-Derived Metabolites Affecting the Immune System

#### 2.1.1. Short Chain Fatty Acids (Propionate, Butyrate, and Acetate)

The most well-known microbiota-related metabolites with immunomodulatory properties are the so-called “short-chain fatty acids” (SCFA), including propionate, butyrate, and acetate. Their production is the result of anaerobic bacterial fermentation of dietary fibers within the intestine and especially within the colon [14]. SCFAs are mainly produced by *Bacteroidetes* and *Firmicutes* phyla, in particular, acetate and propionate from the former and butyrate from the latter [14,15]. SCFAs average concentrations are 70–140 mM in the proximal colon versus 20–70 mM in the distal colon, and vary according to the presence of infection or inflammation [14,15,16] (Figure 1). These variations highlight the significant and rapid absorption of SCFA by colonocytes after their production via passive diffusion or via some transporters, such as MCT (Monocarboxylate Transporter) and SMCT1 (Sodium-coupled Monocarboxylate Transporter 1) [17]. Their primary role after absorption is to serve as an energy substrate for ATP generation in the tricarboxylic acid cycle (TCA), especially for the colonocytes [18]. Hepatocytes are the second main consumers, and only a small part of SCFA reach the systemic circulation, especially for propionate and butyrate [19]. Indeed, the blood concentrations are 25–250 µM for acetate, 1.4–13.4 µM for propionate, and 0.5–14.2 µM for butyrate [20]. At a cellular level, they exert a major role on gene expression through HDAC (Histone DeAcetylase) inhibition and HAT activation (Histone acetyltransferase) [21,22]. SCFA signaling through GPCRs (G protein-coupled receptors), especially three types of GPCRs: FFAR2 (GPR43), FFAR3 (GPR41), and the niacin/butyrate Receptor (GPR109A), which affect chemotaxis, apoptosis, proliferation, and cell differentiation [23,24,25,26,27]. SCFAs are necessary for intestinal homeostasis and for IECs growth [28]. They are also critical for maintaining intestinal barrier integrity through hypoxia-inducible factor 1 alpha (Hif1-α) stabilization [29]. SCFAs can lead to opposite effects on intestinal stem cell growth, depending on their concentration. Lower butyrate concentration in intestinal villi crypts compared to apex appears to inhibit the proliferation of intestinal stem cells through the activation of the transcription factor: FOXO3 [28].

Among SCFAs, butyrate is the main immunomodulatory metabolite, displaying immunosuppressive properties [30]. At the level of adaptive immunity, butyrate promotes regulatory T-cells (Tregs) and IL-22 production through several mechanisms [31,32]. First, by acting as an HDAC inhibitor, butyrate increases FoxP3 expression [33,34]. It also activates indoleamine 2,3 dioxygenase 1 (IDO-1) (that leads to tryptophan deprivation and promotes the production of the immunosuppressive metabolites kynurenine, among others), and aldehyde dehydrogenase (ALDH)1A2 (which is involved in acid retinoic metabolism), two enzymes promoting the conversion of naïve T-cells into FoxP3+ Tregs [33]. Finally, it suppresses the conversion of naïve T-cells into proinflammatory interferon-γ (IFN-γ)-producing cells [33]. It also inhibits Dendritic Cell (DCs) activation/maturation via GPR109A. Butyrate-induced tolerogenic DCs further contribute to immunosuppression through IL-10 production, generation of Tregs, and decrease in T helper (Th)17 [35,36,37]. At the level of innate immunity, butyrate promotes macrophage differentiation, and antimicrobial activity through HDAC3 inhibition that leads to increased ROS production. Hence, butyrate promotes through macrophages the reduction of bacterial translocation, contributing to prevent gut inflammation [38,39]. Due to their antiinflammatory and HDAC inhibitory properties, SCFAs are critical for health. For instance, their properties are implicated in Inflammatory Bowel Diseases, especially in Colitis [40]. Finally, in certain pathophysiological conditions, SCFA mediated immunomodulation is not restricted only to antiinflammatory properties, but it also depends on their concentration, cell type, and metabolic state. Several studies have described the immunostimulatory properties of SCFAs [41]. SCFAs have been shown to increase IL-18 production by IECs, as well as proinflammatory cytokines and chemokines, such as TNF-α (Tumor Necrosis Factor α), IL-6, CXCL1 (C-X-C Motif Chemokine Ligand 1), and CXCL10 by colon epithelial cells in vitro [42,43] (Figure 1). SCFAs improve even the generation of Th1 and Th17 T lymphocytes in an infectious context [44].

#### 2.1.2. Lactate

In addition to the common SCFAs, lactate, which is a hydroxy-carboxylic acid, is derived from food and easily produced by intestinal lactic acid bacteria (LAB), *bifidobacteria*, and other anaerobes [45,46,47] (Figure 1). In order to stabilize the pH and to preserve the balance between the different microbial populations and metabolism of the colonic microbiota, the concentration of lactic acid is closely regulated in the gastrointestinal tract by two mechanisms. The first one is the cycle of production and consumption of lactate, which is used as an energy substrate by other microbiota bacteria. The second one involves the metabolization of lactic acid into SCFAs, especially propionate and butyrate, which contributes to decrease its concentration in the intestinal lumen. Lactate concentration is ranging from 5 to 10 mM in the intestinal lumen and increases in patients suffering from gastrointestinal disorders [48,49]. Microbial communities with low numbers of lactate-utilizing bacteria are inherently less stable, and therefore, more prone to lactate-induced perturbations [50]. Interestingly, on innate immunity, lactate acts as a signaling molecule on GPR81, a cell-surface G-protein-coupled receptor. The GPCR-mediated signaling of metabolites is not covered in the present review, as it has already been extensively reviewed in [51]. GPR81-mediated signaling in colonic DCs and macrophages plays an important role in suppressing colonic inflammation [52]. Lactate also induces downregulation of myeloid cell activation mediated by Lipopolysaccharide (LPS) and leads to a decrease in the production of TNF-α and IL-6 [53]. Moreover, it promotes M2 polarization of macrophages producing a high level of IL-10 and a decrease in the production of IL-12 [54]. Furthermore, Lactate is internalized into the cell by transmembrane proteins, such as MCT-1/4, SLC5A12, and SCL5A8 (subtypes of SMCT channel), and it modulates the gene expression induced through the inhibition of HDAC activity (Figure 1). Lactate is also responsible for metabolic reprogramming, inducing inhibition of glycolysis [55]. In addition to promoting an immunopermissive environment by targeting immune cells present in lamina propria, lactate plays a critical role in regulating IECs inflammatory activity. It inhibits the activation of IECs depending on TLRs and IL-1β. On adaptive immunity, lactate has been shown to decrease the motility of both CD4+ and CD8+ T-cells and the cytolytic activity of CD8+ T-cells [56]. Lactic acid also decreases the proliferation and effective functions of effector T (Teff) cells, and at the same time, increases the conversion of naive to Tregs [57]. Moreover, lactic acid has been implicated in the fight against the development and the aggravation of some pathologies. Lactic acid produced by *lactobacilli* has shown to be essential in regulating inflammation-induced during injury of the small intestine by indomethacin. A metabolomics study has been shown a difference in the concentration of some metabolites in the blood of patients suffering from celiac disease, and notably an increase of the concentration of lactate, which highlights an alteration of the energy metabolism [58]. Lactate may rise to high concentrations (90 mM) in the colonic lumen of patients suffering from Ulcerative Colitis (UC) [48]. In line with this, it has been shown that the presence of lactate triggers *Rorc* and *Il17* expression by naïve and polarized CD4+ T-cells upon in vitro activation. These data suggest that high lactate concentration might promote IL-17-driven inflammation in those diseases.

#### 2.1.3. Succinate

Similar to lactate, succinate, a dicarboxylic acid, is produced in large amounts during bacterial fermentation of dietary fibers [15]. Succinate is a metabolic intermediate of the tricarboxylic acid (TCA) cycle; therefore, it is produced by both bacteria and immune cells. However, as Germ-free (GF) mice have a low amount of succinate in the gut lumen, microbes seem to be the principal producers within the gut [59]. Succinate concentration is very low, ranging from 1 to 3 mM in the intestinal lumen and feces [60]. This is mainly explained by the succinate metabolization into SCFAs, particularly for the production of propionate. Several studies have highlighted the role of succinate as a proinflammatory mediator. In particular, the expression of succinate receptor 1 (SUCNR1, also known as GPR91) on macrophages is associated with an M1 phenotype and a type I response upon stimulation [61]. Upon activation, macrophages change their metabolism to produce and secrete a significant amount of succinate. They simultaneously upregulate SUCNR1, which acts as an autocrine or paracrine sensor for extracellular succinate to subsequently enhance IL-1β production via Hif-1α activation [62]. Hif-1α is a transcription factor that senses and responds to the metabolic state of the cell, acting as a key mediator in mounting both innate and adaptive immune responses [63]. Extracellular succinate also triggers the SUCNR1 receptor expression on DCs, acts as a chemotactic factor, and participates in their activation via TLR3 and TLR7 [64,65]. Finally, it is also reported that succinate is a potent activator of intestinal tuft cells and leads to the activation of type 2 innate lymphoid cells in the lamina propria and the remodeling of the small intestine [66,67,68]. A dysbiosis, including the reduction of succinate-consuming bacteria, can lead to both succinate accumulation and lower SCFAs production within the gut lumen, as observed mainly in patients with a high-fat diet or after antibiotic treatment [69]. Succinate accumulation seems to play a role in acute gut inflammation and may participate in the maintenance of inflammatory bowel disease (IBD) [61,70]. A recent review gathered evidence suggesting that dysbiosis and the imbalance in the SCFA/succinate ratio is one of the causes of IBD [71].

#### 2.1.4. B Vitamins

B vitamins (VB) form a group of hydrosoluble micronutrients counting eight members. Interestingly, though dietary intake is the principal source, the microbiota plays a large part in the daily intake of VB, either by modifying dietary precursors or by de novo synthesis. Most of these modifications cannot be performed by the mammalian host. Genes related to enzymes involved in VB synthesis are scattered across bacterial and yeast genomes [72]. Early genomic research for these pathways in the digestive tract found that half of the bacteria lacks at least one gene for their synthesis [73]. Producing an accessible and active form of VB involves multiple actors among the microbiota, to finally produce the VB benefiting the host [74]. Thus, VB represents an explicit example of a symbiosis both within a polymicrobial community [75], and between the microbiota and the host. As the links between microbes and VB become clearer, the synthesis of each vitamin can now be related to specific bacteria. VB-producing strains evolve through life; for instance, in early childhood, VB2, VB6, and VB9 producing bacteria strains predominate, whereas VB1 and VB5 producing strains are more prevalent in older ages [76]. The discovery of conserved microbial phenotypes in human populations, known as enterotypes, has also led to the findings that enterotypes are specialized in the production of some VB rather than others, without further mechanistic explanation [77,78]. All VB are enzyme cofactors involved in various pathways, ranging from fatty acid metabolism to DNA repair, and their deficiency leads to serious systemic manifestations, which, for most of them, led to their discovery. The indisputable influence of such vitamins on immunity has been extensively reviewed elsewhere [79], but we briefly summarize here the main described VB effects on immunity reviewing the most up-to-date articles (Figure 2).

##### Vitamin B1

Thiamine (VB1) is one of the first VB discovered, due to the particularly notable symptoms of its deficiency, such as Beriberi disease or organ-related diseases, mostly in the central nervous system, as seen in Wernicke’s encephalopathy or optic nerve disorders. VB1 participates in aerobic cellular energy production through amino-acids and alpha keto-acids catabolism, and its derivative, thiamine pyrophosphate (TPP), acts as a cofactor in pyruvate dehydrogenase and the TCA cycle. VB1 comes from diet, and microflora in the large intestine is easily internalized thanks to high-affinity thiamine transporter-1 and -2 (SLC19A2 and SLC19A3) [80]. The TPP, the active form of VB1, which is transformed by the gut bacteria, is an important cofactor for enzymes, such as pyruvate dehydrogenase, alpha-ketoglutarate dehydrogenase, and transketolase [81]. Finally, many factors influence thiamine absorption, such as alcoholism and digestive pathogens, but also multiple other factors independent of the microbiota [82,83]. From the immunological standpoint, VB1 plasmatic concentrations seem to be inversely linked to immune cell counts (Figure 2). Malignant lymphocytic proliferations have been correlated to a higher risk of VB1 deficiency, probably due to intense VB1 consumption [84]. Inversely, patients with low CD4^+^ T-cells in HIV seem to have higher VB1 blood concentrations, even in the early stages of the disease [85], suggesting VB1 consumption requirement for lymphocyte production, activation, or survival, as demonstrated in other animals [86].

##### Vitamin B2

Riboflavin (VB2) is the precursor for two coenzymes, the flavin mononucleotide (FMN) and flavin adenine dinucleotide (FAD), that are required, among others, in the metabolism of niacin (VB3) and pyridoxine (VB6), respectively. VB2 is naturally accessible from a wide variety of foods (e.g., eggs, organ meats, lean meats, and milk are particularly rich in riboflavin). Its production, however, is enhanced by LAB residing in the large intestine, where the VB2 is efficiently absorbed by colonocytes. Riboflavin blood levels correlate with LAB quantity [87], with microbiota diversity and its enterotype [88], as well as the host’s nutritional status [89]. It was shown that VB2 supplementation in mice led to a decrease in proinflammatory cytokines under septic conditions and obesity-associated inflammatory states [90], notably by decreasing TNF-α production by macrophages (Figure 2). VB2 also enhances the macrophage’s production of reactive oxygen species (ROS) and their phagocytic activity [91]. These effects could be a result of NF-κB pathway tampering, or secondary to the proteasome inhibitory properties of VB2 [92]. In addition, human mucosal-associated invariant T (MAIT) cell receptors (TCRs) recognize bacterial riboflavin pathway metabolites through the MHC class 1-related molecule MR1 [93]. MAIT cells kill host cells infected with bacteria and yeast, and secrete soluble mediators, such as TNF-α, IFN-γ, IL-17, etc. The fluctuations of MAIT, both in blood samples and intestinal lumen, are actively studied in the field of IBD [94]. In human diseases, a three week riboflavin supplementation in Crohn patients is beneficial on clinical symptoms, systemic inflammation, and improved the composition of the gut microbiota, underlying its therapeutic potential [95,96].

##### Vitamin B3

Niacin (VB3) is the precursor of nicotinamide adenine dinucleotide (NAD), a crucial electron transporter in central metabolism, oxidative stress control, mitochondrial homeostasis, and senescence regulation. Interestingly, NAD is metabolized through different possible pathways, which often imply transkingdomal cooperation [97], sometimes relying on VB3 intake and sometimes on the kynurenine pathway [98,99]. VB3 directly activates GPCR109A, which also recognizes butyrate, leading to similar effects, notably on lipid regulation. Reintroduction of *Akkermansia muciniphila* in GF mice partially restored the lack of nicotinamide-producing species [100]. Another study suggests the involvement of *Methylobacteriaceae* in VB3 synthesis [101]. VB3 and derivatives are known to participate in macrophage polarization, towards proinflammatory or unconventional phenotypes depending on the stimuli [102]. VB3 also tampers the reactive oxygenated species production both by macrophages and neutrophils, increasing survival in LPS induced sepsis models in mice [103], along with the neutrophils’ myeloperoxidase production and recruitment [104,105]. In preclinical models of colonic inflammation, activation of GPCR109A seems to correlate with the abundance of Treg in the gut lamina propria and the production of IL-10 [37], and with intestinal innate lymphoid cells polarization [106] (Figure 2). 

##### Vitamin B5

When considering pantothenic acid (VB5), evidence is scarcer. Because of its wide distribution in nutritional sources, deficiency in pantothenic acid is very uncommon. However, its relevance in cell metabolism is indisputable, as a precursor of the coenzyme CoA and as a catalyzer for TCA mediated oxidation reactions in fatty acids metabolism. Pantothenic acid has been linked to inflammatory homeostasis. Firstly, metabolic pathways allowing this vitamin to be synthesized into coenzyme CoA are strongly expressed in white blood cells, and the tampering of these pathways leads to unsolicited inflammatory reactions. Panthotein metabolism is relevant to cellular adhesion and polynuclear efficacy, and oxidative stress control, among others [107]. In both isolated macrophages and infected mice, supplementation of VB5 helped clearance of *Mycobacterium tuberculosis* and promoted antibacterial cytokine production [108]. Finally, Ghosal et al. developed a murine model with a knockout (KO) for the gene coding for the intestinal transporter of VB5 and biotin (VB8) (SLC5A6) and shown histopathological alterations associated with a high mucosal inflammation of the colon of those KO mice [109]. This intestinal inflammation was reduced by supplementing the lacking vitamins to the mice’s diets [110]. 

##### Vitamin B6

Pyridoxine (VB6) and its active derivatives, pyridoxal, and pyridoxamine, assist as cofactors to various reactions (e.g., pyridoxal 5′phosphate in amino acid metabolism, but also carbohydrate and lipid metabolic pathways, kynurenine metabolism, and to VB3 synthesis). Isolated VB6 deficiency is rare, frequently associated with other deficiencies, and it manifests itself by a large array of nonspecific symptoms [111]. As with other VB, the different metabolic pathways required to synthesize the bioactive forms of VB6 are shared by the microbiota [112]. The most notable pathways can be, for example, found between *Bacillus subtilis* and in *Escherichia coli* [113]. Studies on VB6 supplementation in human patients without underlying deficiency failed to strongly link the flora and the VB6 uptake [114]. In carps, a VB6 depleted diet accentuated the transcription of proinflammatory cytokines in all segments of the digestive tract [115]. In patients with rheumatoid arthritis (RA), inflammatory biomarkers, such as TNF-α or C-Reactive Protein, were inversely correlated to VB6 level in sera [116], but again VB6 medication in non-deficient subjects showed no beneficial effect [117]. A similar observation was made in patients with critical illness [118,119]. Curiously, in murine models of IBD with a KO in IL-10, both deficiency and supplementation of VB6 had beneficial results on gut inflammation [120]. VB6 is likely to be involved in regulating the NF-κB pathway [121], and lymphocyte polarization in favor of a Th1 mediated immune response, rather than Th2 [122,123,124]. However, the participation of VB6 derivatives in the kynurenine and sphingosine pathways complicates the understanding of its genuine implications in human health and microbiota. At this time, it seems possible that VB6 is required for a stable, non-inflammatory baseline state, and perhaps to limit the cytokine storm in COVID-19 [125], but further evidence is needed.

##### Vitamin B8

Biotin (VB8) is the vitamin of the B group with the most admitted relevance in cellular health. It catalyzes up to six vital carboxylases involved in numerous roles, ranging from fatty acids to amino-acid metabolism [126]. The understanding of the role of VB8 on immune health, and by extension, the consequences of VB8 deficiency or supplementation, is yet still beginning and has been reviewed elsewhere [127]. Notably, its possible involvement in the NF-κB pathway regulation holds high expectations in a potential antiinflammatory process. Recently, this hypothesis has been strengthened by Skupsky et al., which showed a lowered NF-κB activity and histological signs of mucosal inflammation in the colon of biotin-supplemented mice [128]. This observation concurs with a previously mentioned study, exploring the supplementation of VB5 and VB8 [110]. Another team focused on the properties of VB8 on human T-cell population, and showed that VB8 deficient growth media promoted higher titers of proinflammatory cytokines, such as IFN-γ and IL-17, enhanced the T-cell polarization towards Th1 and Th17 proinflammatory profiles, and reduced the proportion of Treg-cells through the rapamycin-mTOR pathway [129].

##### Vitamin B9 and B12

Folate (VB9) and cyanocobalamin (VB12) are often studied together. In a human randomized trial, dietary uptake of probiotic species (an association of *Bifidobacter* spp. and *lactobacillus* spp.) was associated with higher blood titers of VB9 and VB12, without any additional supplementation of these vitamins [130]. Folate is an essential cofactor for synthesizing amino acids, nitrogenous bases, and ribonucleoside, which is why antifolates are commonly used in cytotoxic chemotherapies and antiinfectious treatments. It is also involved in mitochondrial RNA metabolism [131]. VB12 also catalyzes the synthesis of amino acids, both from the cell and the mitochondria [132]. Evidence of bacterial involvement in their synthesis and absorption is indisputable since their full metabolic pathways have been elucidated [133,134,135,136,137]. In immune homeostasis, though both vitamins are required for hematopoiesis, the role of VB9 is better understood. Lower titers of VB12 seem to correlate with higher inflammation levels in critically ill patients [138], and were associated with higher mortality in hospitalized patients [139], but the vitamin’s direct involvement remains unclear. VB9 titers, on the other hand, are strongly correlated with leukocyte counts [140]. VB9 deficiency induces DNA damage in lymphocyte precursors, similarly to those observed in radiation-induced injuries [141]. In allergic children, excessive levels of VB9 are also proportional to allergic symptoms, and inversely proportionate to circulating regulatory T lymphocytes levels [142]. Folate-associated receptors are found on mature immune cells, e.g., naive T lymphocytes, and might be involved in their activation or regulation [143]. Similar receptors on the surface of macrophages are overexpressed under certain conditions, such as hyperlipidemia, and lead to activation of macrophages in a proinflammatory profile, promoting, for example, atherosclerosis [144]. These receptors are targets for potential antiinflammatory drugs, notably for the control of chronic inflammatory states [145] and chronic IBD [146].

#### 2.1.5. Amino Acids (AA) and AA-Derived Metabolites

##### Tryptophan

Tryptophan (Trp) is an essential amino acid mainly supplied by the diet. The majority of Trp is absorbed in the small intestine. A small portion can reach the colon, where it can be used by commensal bacteria. In humans, three Trp pathways are generally described: The kynurenine pathway, the indole pathway, and the serotonin pathway. Serotonin (also known as hydroxytryptamine or 5-HT) can modulate the immune response, and hence, potentially influence intestinal inflammation [147]. It has been demonstrated that GF mice increased plasma serotonin concentrations, which can be normalized following colonization of the mice immediately post-weaning [148]. Several of the 5-HT receptors have been associated with immune cells, such as lymphocytes, monocytes, macrophages, and DCs, which indicates that 5-HT plays an immune-modulatory role. Tryptamine concentrations increase nearly 200 fold in feces following colonization of GF mice with human gut microbiota, suggesting that bacterial metabolism of Trp generates luminal tryptamine [149]. The enzyme tryptophan decarboxylase (TDC1) is responsible for converting Trp to tryptamine, which is converted to serotonin by tryptamine 5-hydroxylase. The genes encoding the homologs of this TDC1 were found in at least 10% of the representative human gut microbiota [150]. Two comprehensive reviews have already discussed the link between the gut microbiota regulation of Trp and its impact on health and disease [151,152]. In addition to tryptamine, other metabolites, like indole derivatives or skatoles, are derived from the catabolism of Trp by the intestinal microbiota. Those are described in the review [153]. They have been intensively studied for their role in immunity, mainly through the activation of the aryl hydrocarbon receptor (AhR). In addition to this activity, indole is now studied for its ability to influence the gene transcription in some immune cell subsets and epithelial cells. Indeed, Microbiota-derived AhR ligands play a critical role in gut homeostasis; for instance, they have been associated with the increase of *Il22* transcript in type 3 innate lymphoid cells or ILC3 [154], and subsequent mucosal integrity. Trp-derived indoles also activate the pregnane X receptor (PXR) to support the barrier function in colitis mouse models [155]. One example is indole 3-propionic acid (IPA) produced by *Clostridium sporogenes*. This metabolite is absent in GF mice. The ability of *C. sporogenes* to produce IPA from Trp was recently described and is limited to a small group of organisms sharing the proper metabolic pathway [156]. GF-mice colonization with an IPA-producing strain like *C. sporogenes* restores the IPA serum level [13]. It was shown to activate the AhR and the PXR pathways leading to the reduction of the symptoms in a mice-dextran sodium sulfate (DSS) model of colitis [157,158]. Thus, IPA might be an interesting candidate for the treatment of UC. Other indoles, such as Indole Acrylic Acid (IA) [159] and Indole-3-aldehyde [154], are described in the literature for their effects on immune cells (Figure 3). Microbial metabolism of Trp resulting in indole derivatives, such as indole acetic acid and IPA, has been recently investigated in several studies for their contributions to host physiology. In the future, Trp-derived metabolites should be considered as a promising strategy that can be used to treat human diseases [160].

##### Taurine

Taurine is a sulfur-containing amino acid that can be sourced from the diet or synthesized by the pancreas via the cysteine sulfinic acid pathway. Taurine is not per se a metabolite produced by bacteria. However, as described below, taurine reaches the colon in a conjugated form with bile acids (Bas). Its release in the colon depends directly on microbiota activity that can result in taurine deconjugation via bile salt hydrolase (BSH) activity. Thus, it can be considered as an indirect product of the colonic microbes. Taurine is known to be involved in many physiological events, such as osmoregulation, membrane stabilization, calcium mobilization, neurotransmission, reproduction, and detoxification [161,162,163], along with providing antiinflammatory effects and protection to cells from cytotoxic effects of inflammation. During Bas synthesis, glycine is more often used than taurine (ratio of 3:1) [164]. However, this ratio is subject to variation since the availability of taurine is diet-dependent (a western diet will favorize a taurine conjugation when vegetarians are using glycine [165,166]. In the Dexamethasone-induced immunosuppressive mice model, taurine was involved in the increase of lymphocytes in Peyer’s patches, a well-known lymphoid follicle in the intestine classified as a gut-associated lymphoid tissue [167]. Similarly, changes in the microbiota were observed after treating the mice with taurine. These groups showed a modification of the pattern, suggesting that some beneficial bacteria, such as *Lachnospiraceae* and *Ruminococcaceae*, groups were significantly re-increased after treatment with taurine. The role of taurine on gut epithelial cells was also presented by Levy et al. and reported as an activator of the NLRP6 inflammasome leading to the activation of IL-18 and the production of antimicrobial peptides (AMPs) [168].

##### p-Cresol and Its Derivatives

p-Cresol is a methyl phenol produced by colonic fermentation of tyrosine and phenylalanine. After its production, most of the p-cresol (approximately 80%) in the intestinal wall is conjugated by the intestinal flora into p-cresyl sulfate (pCS) or p-cresyl glucuronidate. The association between the microbiota and p-cresol or its derivatives has been proven, for example, in GF-mice models, in which p-cresol or its derivatives were not found in the blood [13]. In humans, pCS is considered a uremic solute, and patients with chronic kidney failure display high blood pCS levels. However, among these patients, those who are also colonectomized normal display levels of pCS, demonstrating the colonic origin of pCS synthesis [169]. Culture-based tests have identified several intestinal bacteria as producers of p-cresol, such as those belonging to the *Coriobacteriaceae* and *Clostridium clusters* XI and XIVa [170]. A comprehensive overview of bacterial species able to produce phenolic compounds has been reviewed elsewhere [171]. Derivatives of p-cresol are considered as uremic toxins, involved in uremic syndromes developed during renal failure progression. The immunomodulatory effect of p-cresol and derivatives has been highlighted in chronic kidney disease (CKD) patients that are highly susceptible to infectious diseases [172]. The most studied p-cresol derivative is pCS, for its role on both innate and adaptive immunity. In an in vitro study, it was found to induce ROS production and phagocytosis at lower concentrations, while at higher concentrations, it was shown to interfere with antigen processing in human monocyte-derived macrophages [173]. Moreover, in a mouse model of adenine-induced renal dysfunction, pCS was found to be involved in immune dysfunction in CKD. It decreased peripheral B lymphocytes number by inhibiting the proliferation of CD43+ B-cell progenitor [174]. The role of pCS on Th1-type immune responses has been studied through a tyrosine-enriched diet mouse model. pCS was negatively correlated with the percentage of IFN-γ-producing Th1 cells during a 2, 4-dinitrofluorobenzen-induced contact hypersensitivity. In vitro assays on splenocytes exposed to a variable concentration of pCS suggested that intestinal-derived pCS suppresses the percentage of IFN-γ-producing Th1 cells and favors a Th2 response [175]. Finally, while the role of p-cresol derivatives has been increasingly characterized, the exact effect of p-cresol itself is still controversial [176]. Even if CKD is a good study model, it may be interesting to understand if pCS can influence the immune system in a healthy state, as p-cresol derivatives are constantly eliminated in urine through tubular secretion.

##### Histidine and Derivatives

Gut bacteria can convert amino acids, such as L-histidine, into biogenic amines (i.e., molecules containing one or more amino groups), such as histamine, by the action of histamine decarboxylase (HDC) [177]. Numerous immune cells produce histamine, mainly basophilic cells, and mast cells, and to a lesser extent, monocytes, DC, and lymphocytes [178]. Histamine production by basophils and mast cells, is modulated by cytokines, such as IL-3, IL-12, IL-18, TNF-α [179]. Histamine is mainly produced within the caecum, and found at lower levels in the wall of the small intestine in GF mice, compared to conventional mice [180]. Histamine, in addition to its role as a neuromediator and regulation of gastric acidity, is mainly described as a cytokine and an inflammatory mediator in acute inflammation and hypersensitivity. Recently, it has been shown that histamine produced by microbiota-associated bacteria impacts both intestinal epithelial cells and immune cells. In humans, Barcik et al. have performed a PCR analysis of bacterial HDC expression on fecal samples from 74 healthy donors, and identified *E. coli*, *Lactobacillus vaginalis*, and *Morganella morganii* as histamine-secreting bacteria. Interestingly, they also collected samples from 74 asthma patients, highlighting an increased bacterial HDC copy number compared to healthy donors. Furthermore, *Morganella morganii* relative abundance was correlated with disease severity, suggesting a possible involvement in asthma pathogenesis [181]. Histamine action is mediated by four types of receptors noted H1R to H4R, and leads to different effects, depending on the receptor subtype, its expression level, as well as the targeted cell. DCs express H1R, H2R, and H4R, receptor subtype. H1R engagement is involved in immediate hypersensitivity reaction, inducing vasodilatation, and promoting Th1 responses, while H2R activation leads to an inhibition of Th1 and Th2 responses and promotes Treg by increasing the production of IL-10 [182,183,184]. On adaptive immunity, in the same way, H1R promotes IFN-γ, the production, and the proliferation of Th1 cells, while H2R inhibits Th2 response by blocking the synthesis of IL-4 and IL-13 [178,185]. Thus, histamine affects the balance between Th1 and Th2 responses [178]. Levy et al. analyzed the immunomodulatory effects of some microbiota-associated metabolites and showed that histamine intake in drinking water led to a strong reduction of IL-18 proinflammatory cytokine production by inhibiting NLRP6 inflammasome assembly [168]. Of note, the therapeutic potential of a bioengineered histamine-overexpressing bacteria has also been investigated in an in vivo asthma model. When orally administrated, it led to a decrease of Th2 cytokines secretion by lung-derived cells, as well as proinflammatory cytokines and IL-10. These results suggest a global antiinflammatory effect, not only mediated by an inhibition of Th2 response. Thus, these data indicate that histamine-producing bacteria within the gut can potently modulate systemic host immunity [186].

##### Polyamines

Among biogenic amines, polyamines (Pas) are organic compounds having more than two amino groups. They have been studied for their ability to modulate cellular functions, including gene regulation, stress resistance, cell proliferation, and differentiation [187]. Spermidine (N3) and spermine (N4), two putrescine-derivatives, are the major examples of Pas in human cells. These two metabolites can be found in food or directly produced by gut microbiota or any eukaryotic cells. While the ingested food is the major source of Pas in the small intestine and is a crucial source for the stability and the maintenance of gut tissues [188], the microbiota appears to be a major player in Pas production in the lower part of the intestine, i.e., the colon [189,190,191]. Pas are extensively studied for their interaction with epithelial cells and the immune system. Interestingly, Levy et al. reported that exogenous spermine significantly reduces the activation of NLRP6 inflammasome in IECs and subsequent release of IL-18, a cytokine that promotes the production of antimicrobial peptides (AMPs), an important element involved in gut microbiota modulation [168]. Spermine also exerts antiinflammatory effects directly on immune cells by inhibiting LPS-induced expression of proinflammatory cytokines, including IL-6, TNF-α, IL-1, MIP-1α, MIP-1β, by monocytes and macrophages [192,193]. This was confirmed in an in vivo sepsis mouse model where a systemic antiinflammatory effect was observed [194]. In addition to spermine, spermidine is also of interest for its effect in age-associated diseases [195,196,197], cardioprotection [198], tumor suppression, immune modulation, neuroprotection, metabolic syndromes, and stem cell function. All of these aspects have been extensively reviewed elsewhere [199]. Recently, spermidine was shown to play a major role in regulating T-cell differentiation and function. Spermidine exposition potentiates the in vitro Foxp3 + T-cell differentiation from both naïve and Th17 CD4+T-cells, in an autophagy-dependent manner [200], and dampens the IFN-γ-mediated monocyte response [201]. Moreover, the proTreg effect of spermidine has been confirmed in an in vivo colitis mouse model [200]. A considerable amount of studies suggested that Pas have real health benefits, making them potential therapeutic candidates [202,203].

##### D-Amino Acids

While mammalian cells can produce only two D-amino acids (D-AA), namely, D-serine and D-aspartate [204], pathogenic and commensal bacteria can produce several D-amino acids, which play an essential role in multiple biological processes [205]. Sasabe et al. reported that the mouse intestine is rich in free D-AA produced by gut microbiota [206]. They have found the D-amino acid oxidase (DAO), which is associated with D-AA, expressed in the villous epithelium of the small intestine. The oxidative deamination of the intestinal D-AA by DAO results in the generation of hydrogen peroxide, H_2_O_2_, a strong oxidizing agent with antimicrobial activity, and leads to the modification of the human microbiota, protecting the mucosal surface of the small intestine against patens. Thus, they proposed bacterial D-AA acids and DAO as new examples of inter-kingdom communication. Beyond bacterial communication, D-amino acids display antimicrobial properties, such as ROS production, and bacteriostatic activity. They can regulate neutrophil chemotaxis and modulate immune tolerance. These properties have been comprehensively reviewed elsewhere [207].

##### Gamma-Aminobutyric Acid

The gut microbiota has a critical role in the production of gamma-aminobutyric acid (GABA). It is mainly produced through a one-step reaction from glutamate, catalyzed by glutamic acid decarboxylase [208]. It can also be produced by some microorganisms from putrescine [209] (see section polyamines). GABA production was described for several *Lactobacillus* and *Bifidobacterium* strains [210], and it was reported that the introduction of a GABA-producing Bifidobacterium strain was sufficient to modulate GABA levels in the gut [211]. Among them, *Lact. Brevis* and *Bif. Dentium* was those producing the highest amounts of GABA [210]. GABA produced by *Lactobacillus rhamnosus* JB-1 in the gut has been shown to have effects on the brain (the microbiota-gut-brain axis), and, in particular, resulting in a reduction of stress-induced corticosterone, and in depression- and anxiety-related behavior. In the context of enterotoxigenic *E. coli* (ETEC) infection, ETEC induces dysbiosis, increasing the GABA-producing *Lactobacillus lactis subsp. lactis*. The increased content of GABA in the jejunum of mice promotes intestinal IL-17 expression through a GABA-mTORC1 signaling pathway [212]. The effects of GABA on intestinal homeostasis have been described through its action on enterocytes. Firstly, it has been described as a selective stimulator of mucin-1 expression in epithelial cells [213]. Then, exposure of GABA to epithelial cells resulted in a decrease in IL-1β-mediated inflammation and an increase in tight junctions and transforming growth factor beta (TGF-β) expression, thus providing a protective effect against the disruption of the intestinal barrier [214].

##### Quorum Sensing Molecules

Quorum sensing molecules (QSMs) consist of signaling molecules, sensing molecules, and downstream regulatory proteins produced and used by prokaryotes to monitor population density and assist a number of biological functions, such as biofilm formation and production of virulence factors. The gut microbiota has shown to have a very stable structure despite challenges in the gut, such as stomach acid and intestinal bile; its stability depends on QSMs [215]. Generally, in Gram-Negative bacteria, quorum sensing (QS) is composed of two components: A small soluble signaling molecule (predominantly N-acyl homoserine lactone (AHL) molecules) and a transcriptional regulatory protein (R protein). In gram-positive bacteria, QS systems are generally composed of three components, a signal peptide and a two-component regulatory system (TCS) itself composed of a membrane-bound histidine kinase (HK) sensor and an intracellular response regulator (RR) [216,217]. The main QSMs are AHLs, diketopiperazines (DKPs), 4-hydroxy-2-alkylquinolines (HAQs), diffusible signal factors (DSFs), autoinducer-2 (AI-2). QSMs are products of nutrients utilization, and many of them are synthesized from metabolites present in the environment. For instance, AHLs and AI-2 are both produced from S-adenosylmethionine, which is part of the methionine metabolism. AHLs are characterized by carbon acyl chains, some of which are derived from intermediate molecules of the fatty acid biosynthesis in the host [216]. Surprisingly, some organisms use more than one QS signaling system, which means that QS is not only involved in intra-species communication, but also involved in inter-species communications. It is the case of AI-2, which is used both by Gram-positive and Gram-negative bacteria, and which is the only major type of QSM known to promote inter-species bacterial communication across distantly related bacterial species [218]. AI-2 has been shown to influence the composition of antibiotic-treated microbiota and modulate the abundance of certain phyla [219]. QSMs are also detected by immune cells and regulate the activity of both innate and adaptive immunity. For instance, bacterial autoinducers (Ais) elicit proinflammatory effects and modulate the activities of gut-associated T lymphocytes, macrophages, DCs, and neutrophils [220]. Moreover, Mast Cell activation through the mast-cell-specific GPCR, MRGPRX2, which are Gram-positive QSMs receptors [221] leads to degranulation and release of ROS, TNF-α, and Prostaglandin D2, thereby triggering a local inflammatory response. In 2019, the association between gut microbiota-derived AI-2 and the progression of colorectal cancer (CRC) has been highlighted [222]. This association was mediated by tumor-associated immune cells, such as macrophages. AI-2 was proposed as a novel biomarker for human CRC as its concentration was higher in colorectal tissue and stool of CRC groups compared with normal colonic mucosa and colorectal adenoma. Moreover, AI-2 levels increased depending on CRC staging in both tumor tissues and stool samples. A positive correlation was also found between AI-2 and Tumor necrosis factor ligand superfamily member 9 (TNFSF9), a costimulator of T-cell proliferation belonging to the TNF receptor family. This association was confirmed experimentally through the stimulation of macrophages with AI-2 extracts from *F. nucleatum* that showed a significant increase in TNFSF9 expression, thus confirming the link between microbial QS, inflammation, and CRC.

#### 2.1.6. Catecholamines

Catecholamines (Cas), which include epinephrine (E, also known as adrenaline), norepinephrine (NE, also called noradrenaline), and dopamine, are a class of organic compounds characterized by a catechol structure, i.e., a benzene ring bearing two adjacent hydroxyl groups, and a side-chain amine, which contributes to receptor specificity [223]. Cas synthesis starts with dietary L-dopa as substrate, which is then enzymatically converted into dopamine, NE, and finally, E. Cas, especially NE and dopamine, are abundantly present within the human gastrointestinal tract [224]; in particular, approximately half of the NE amount in the mammalian body is produced within the pre-vertebral ganglia innervating the gut mucosa, while dopamine is synthesized in non-sympathetic enteric neurons within the intestinal wall [225,226]. It is unlikely instead to normally find E in the gut at a significant level, since the enzymes required for its conversion from NE are not expressed in the intestinal mucosa [227]. In addition to this, there is increasing evidence that the intestinal lumen Cas levels are gut microbiota-dependent [228]. While data on the magnitude of the microbial contribution to Cas levels in the gut are still scarce, because the normal NE and dopamine luminal concentrations are difficult to determine, it is instead clear that the presence of bacterially derived NE and dopamine contributes to make the gut a Cas-rich environment, together with dietary intake and enteric nervous system sources [228]. Among others [229,230], Asano and coworkers demonstrated that the gut microbiota plays a crucial role in producing biologically active free Cas in the gut lumen [231]. They showed that levels of NE and dopamine increased in specific pathogen-free mice with normal gut microbiota, when bacteria were present. The role of Cas as chemical neurotransmitters in the central and peripheral nervous systems, with a key position in regulating various physiological processes and functions, such as cognitive abilities and intestinal motility, is well-recognized [232]. In addition to this, Cas have shown to be potential inter-kingdom signaling molecules in the gut: In the 1990s, the pioneering work of Lyte et al. demonstrated that some pathogenic species could recognize exogenous Cas in vitro, leading to an increase in bacterial growth [233,234]. Later, Sperandio et al. showed for *E. coli O157:H7* that the two-component regulator sensor kinase QseC is a receptor for the host E and NE [235], supporting the observation that Cas could act as inter-kingdom signals [236]. In *Salmonella Typhimurium*, NE has been shown to trigger the expression of virulence-associated factors, including flagella-mediated motility and Type III protein secretion [237,238,239]. Signaling between host cells and microbes through Cas continues to attract interest among the scientific community, especially as it might play a key role in microbial dysbiosis and increase susceptibility to infection by altering the growth and virulence of human pathogens [228]. However, further investigation is needed to fully elucidate the luminal CA-related functions and their mechanisms. Finally, there are many studies indicating that Cas is also important immunomodulators during health and disease [240,241,242,243]. These functions were first observed in 1904, when pronounced leukocytosis was reported following subcutaneous administration of E in humans. It is now clear that lymphocytes and phagocytes are also Cas-producing cells [244], and there is evidence that Cas exert autoregulatory functions on immune cells by means of intracellular oxidative mechanisms [245]. To conclude, the consequences of lymphocyte/phagocyte-derived or exogenously administered Cas during shock and trauma are becoming increasingly clear [244]. During severe tissue trauma, the body experiences the destruction of noradrenergic nerve cell innervation, with a consequent release of NE into the systemic circulation. Lyte and Bailey reported that this led to increased proliferation of bacteria within the gastrointestinal system of an experimental murine model, probably contributing to the high incidence of systemic bacterial inflammation and sepsis following trauma hemorrhage [246]. Metabolites shared between different cell types and the gut microbiota, such as Cas, are particularly difficult to study. Further research is needed to understand the dynamics of Cas in the gut lumen and its impact on mucosal immunity. Mono-colonization with CA-producing microbial species and well-defined in vivo models will certainly help to decipher the role of each player.

#### 2.1.7. Cyclic-Dinucleotides (CDNs) and Cyclic-Trinucleotides (CTNs)

CDNs are small nucleic acids produced by bacteria (and viruses) that act as essential secondary messengers that help them to coordinate a new response and shape their behavior after a change in their environment. For example, cyclic di-adenosine monophosphate (cyclic di-AMP) is involved in bacterial growth, biofilm formation, stress response, and antibiotic resistance [247]. When CDNs are sensed by immune cells, this results in proinflammatory responses. Depending on CDNs types, they can activate either STING (Stimulator of interferon genes) pathway, or the oxidoreductase RECON (NF-kB controlling reductase). CDNs with two purine bases (such as c-di-GMP, 3′,3′ cGAMP, and c-di-AMP) triggers STING-mediated NF-κB activation and type-I interferon [248]. In addition, the oxidoreductase RECON can specifically bind to cGAMP, and c-di-AMP, cUMP-AMP, resulting in the inhibition of its enzymatic activity and promoting a proinflammatory antibacterial state [249]. The RECON specificity is not limited to purine bases. It can also sense cyclic-trinucleotides (CTNs), such as cAMP-AMP-GMP produced by bacteria. Some of these CTNs are only recognized by the RECON sensor, but not by STING, leading to the inhibition of the NF-κB responses [250]. In a pathological context where intracellular bacteria are involved, the ability of these bacteria to increase the diversity of their CDNs/CTNs can be considered as an evolutionary adaptation mechanism to evade host innate immunity and metabolic pathways. This area of research is still emerging, and we are certainly at the beginning of a concept.

#### 2.1.8. Inosine

Inosine is a nucleoside formed from ribofuranose and hypoxanthine. Recently, it was shown that inosine produced by *Bifidobacterium pseudolongum* is able to change the immune response after an immune checkpoint blockade (ICB) therapy [251]. In their work, Mager et al. studied the difference in microbiota between CRC-positive mice challenged with conventional immunotherapy versus CRC-positive mice with no treatment. Seven bacterial species were found to be specific to ICB-treated tumors. A mono-colonization assay with five of the seven species in a heterotopic model of CRC showed that only *B. pseudolongum* had the most important effect on immunotherapy, and this effect was mediated by inosine, which is found in high amounts in the sera of mice mono-colonized by this bacteria. Inosine can activate the adenosine 2A receptor in immune T-cells, and drive the Th1 differentiation of costimulated T lymphocytes.

#### 2.1.9. Secondary Bile Acids

Bile acids are hydroxylated steroids involved in digestion and lipids absorption; they are considered hormones. Currently, it is known that primary bile acids, such as cholic acid and chenodeoxycholic acid, are synthesized in the liver and secreted in the intestine in a conjugated form with glycine or taurine. 95% of them are finally reabsorbed in the ileum and recycled by the liver; this cycle is known as enterohepatic circulation. The remaining 5% activates the FXR in the liver and intestine tissues. This leads to the repression of both the CYP7A1 gene, encoding for a rate-limiting enzyme in the classic bile acid synthesis pathway, and the CYP8B1 gene, required for cholic acid synthesis [252]. The microbiota interacts with the metabolism of bile acids at different stages. First, primary bile acids are deconjugated by bacterial communities via BSH. Then, they can also be transformed by the microbiota to produce more than 20 secondary bile acids (e.g., deoxycholic acid (DCA) and lithocholic acid (LCA)). Secondary bile acids can act as soluble mediators signaling through bile acid receptors (BARs) expressed by many types of cells, such as epithelial cells and some immune cell subsets. BARs encompassed a large family of GPCRs (e.g., GPBAR1) and nuclear (e.g., FXR) receptors. BARs can recognize both primary and secondary bile acids. A given BAR can recognize several bile acids that can either trigger an activating or an inhibitory signal, depending on the bile acid and BAR involved (Reviewed in [253]). They are viewed as a negative regulator of macrophages [254], DCs, and NKT [255] functions, supporting the idea that they contribute to the maintenance of a tolerogenic environment in the liver and intestine [256]. Recently, Song et al. demonstrated that deconjugated bile acids can promote the generation of colonic RORγt-expressing FoxP3+ Tregs through Vitamin D receptor (VDR) [257]. They demonstrated that genetic abolitions of bile acids metabolic pathways (required for secondary bile acids production) in individual gut symbiont (*Bacteroides thetaiotaomicron* and *Bacteroides fragilis*) decreases this T-cell population. This effect was cell-type- and tissue-type- specific, as no similar effects were observed either on colonic Th17, nor on Tregs and Th17 from the spleen, the mesenteric lymph node, or the small intestine, respectively. Importantly, these cells have been shown to display superior suppressive capacities in T-cell-mediated intestinal inflammation [258] than conventional FoxP3+ Tregs. Recently, Campbell et al. screened several secondary bile acids and identified the 3β-hydroxydeoxycholic acid (isoDCA) as an immunosuppressive bile acid. IsoDCA alters DC-mediated T-cell stimulation, by decreasing DC costimulatory molecules expression and proinflammatory cytokine production that in turn promotes FoxP3+ Tregs generation [259].

### 2.2. Microbial Membrane Metabolites Affecting the Immune System

Microbes not only modulate host immunity through soluble metabolite secretion, but can also use membrane-bound metabolites. Unlike soluble metabolites, membrane-bound metabolites require more complex mechanisms to cross the epithelial gut barrier. It has been proposed that these strategies are based on outer membrane vesicle (OMV) formation or dead bacteria-derived membrane fragments. Presumably, followed by endocytosis or passive diffusion through epithelial cells, this class of metabolites then interacts with the gut-associated immune system. The entire mechanism needs to be further investigated to clearly understand how these metabolites move from the microbial membrane to immune cells (Figure 4).

#### 2.2.1. Sphingolipids

Sphingolipids are a class of plasma-membrane-associated lipids containing a backbone of sphingosine. They are produced by both the host and specific bacteria. Host sphingolipids participate in a large and diverse range of physiologic cellular functions, as they are involved in specific signaling pathways, either acting themselves as signaling molecules or regulating downstream signaling molecule functions [260]. Within the gut, host-sphingolipid complexes, such as sphingomyelin (SM) and glycosphingolipids (GSLs), are essential components of epithelial cells, playing a role in lipid absorption, providing protection, and integrity of the mucosa. Host sphingolipids (notably ceramide-1-phosphate (C1P), sphingosine-1-phosphate (S1P), and ceramide (Cer)) have been described alternatively as anti or proinflammatory and act directly or indirectly as inflammatory mediators [261]. Along the intestine, the ratio of sphingolipids (pro/antiinflammatory) is tightly controlled, and dysregulation of this balanced ratio is observed in diseases, and notably in IBD. It has been shown that sphingolipid levels are significantly different between inflamed and non-inflamed intestinal tissues and may affect the immune gut ecosystem [262]. The production of SM-like sphingophospholipids is described in the gut from members of the Bacteroidetes phylum (e.g., Bacteroides ([263] and Prevotella [264]). Sphingolipids in sphingolipids-producer bacterial strains participate in stress resistance by preserving membrane integrity [265]. It is believed that Bacteroides-derived sphingolipids may cross the epithelium barrier and reach immune cells via OMV formation [266]. Sphingolipids are presented by antigen-presenting cells (APCs) through the CD1d receptor, an MHC class I-like receptor to CD1d-restricted innate-like lymphocytes called invariant Natural killer T (iNKT). The best known CD1d-restricted iNKT ligand is aGalCer produced by Sphingomonas spp [267]. Recently, Kinjo et al. described that Bacteroides fragilis produced natural aGalCer analogs [267]. Early exposition to microbiota-derived sphingolipids, and notably Bacteroides fragilis-derived α-Galactosylceramide GSL-Bf717, prevents excessive colonic iNKT accumulation [268], and subsequent iNKT-related inflammatory disorders in adulthood [269]. The antiinflammatory properties of Bacteroides-derived sphingolipids were revealed by Brown et al., who showed that patients with Crohn’s disease and UC had elevated levels of host-derived sphingolipids, while the abundance of Bacteroides was decreased [270]. To better understand the exact role of Bacteroides-derived sphingolipids, a GF mice mono-colonization experiment was performed with sphingolipid-deficient B. thetaiotaomicron strain and compared to the wild-type strain. The sphingolipid-deficient strain triggered intestinal inflammation, as revealed by histopathological inflammatory signatures, also characterized by an increase of IL-6 and MCP-1 proinflammatory cytokines within the colon [270]. Interestingly, Ultra-High Performance Liquid Chromatography (U-HLPC) coupled to an Exactive Plus orbitrap mass spectrometer lipidomics analysis revealed 35 unique sphingolipids derived from Bacteroides, highlighting the diversity of these molecules [270]. In conclusion, it is now clear that sphingolipids can modulate host immunity. However, depending on the producing organism, sphingolipids can differentially impact the immunity either toward inflammation or tolerance. Thus, microbial-derived sphingolipids have to be identified, isolated, and screened for their immune properties. 

#### 2.2.2. Lipoteichoic Acids

Lipoteichoic acids (LTA, from the Greek “teichos” meaning wall) are amphipathic polymeric molecules anchored onto the Gram-positive bacteria wall, which regulate several autolytic wall enzymes (muramidases) [271]. They are composed of a repeating unit of glycerophosphate or ribitolphosphate bound to a glycolipid anchor [272]. Another form of bacterial cell wall polymers are wall teichoic acids (WTA), which are instead covalently bound onto the peptidoglycan and that have been reviewed elsewhere [273]. For Gram-positive bacteria, TLR2 ligands are the primary host Toll-like receptors (TLRs) involved in the innate immune response [274]. Even if their nature is still controversial, several studies have suggested that LTA are their main activator [275,276,277]. It appears that LTA can play an immunostimulatory role, by activating in vitro the NF-κB pathways in monocytes and macrophages, in a TLR2-dependent manner [276,278,279]. Other pattern recognition receptors (PRRs), such as C-lectins or Ig-receptors and CD14, have been shown to be involved in the immune response triggered by LTA during bacterial infections [280,281]. The activation of phagocytes promotes the production of cytokines, participating in appropriate antibacterial immune response in Gram-positive bacterial infection models [282,283,284]. However, some of the most compelling evidence on probiotics revolves around the LAB. The LTA of some LAB seems to exert a particular immuno-modulatory effect [285]. In murine models, dietary Lactobacillus rhamnosus allowed better healing from intestinal radiation, and in vitro studies confirmed the involvement of its specific LTA in the protective effect [286]. Other teams have shown a diminution in proinflammatory cytokine production from macrophages when exposed to the same LAB [287]. Animal experimentations with heat-killed Lactobacillus paracasei suggest that the exposition alone to the cell wall is enough to trigger the immunomodulatory effect, along with other beneficial effects of LTA, such as an increase in mucosal production via MUC2 production [288]. Grangette et al. demonstrated the possibility to enhance its antiinflammatory capacity by modifying the composition of an already beneficial type of LTA from Lactobacillus plantarum, therefore emphasizing the role of structural variations of LTA on the modulation of the host immune response [289]. The interactions between LTA and the immune system are not yet fully understood. Different levels of expression for TLR2 as a protective mechanism or a risk factor for inappropriate inflammation, as suggested by Melmed et al. [290], and other complex crosstalks should be furthermore examined to get a grasp on the intricated interactions that developed between bacterial LTA and human immune system during our long coevolution. The structural diversity of LTA and their effects, combined with the complexity of microbiota composition, could participate in several reactions in the immune response, and thus, should be explored in appropriate samples whenever the question is relevant. 

## 3. Metabolites Derived from Immune Cells Affecting the Microbiota

Although interactions between microbiota and host are bidirectional, most studies have mainly focused on the impact of bacteria strains and derived-metabolites onto immunity. To our knowledge, no study has yet described immune-derived “bona fide” metabolites impacting gut microbiota composition and functions. However, some studies highlighted that host immune activation triggers rapid microbiota modulation, at least at the transcriptional (and thus, functional) level. Moreover, it is now clear that the immune system acts on microbiota through several effector molecules, ranging from antimicrobial peptides to soluble Immunoglobulin A (sIgA). In order to emphasize the importance of this bidirectional dialogue, we briefly describe in this part the impact of the host immune system on gut microbiota modulation.

### 3.1. Gut Microbiota Modulation upon Acute and Chronic Host Immune System Activation

Most chronic inflammatory diseases are associated with dysbiotic gut microbiota. Many efforts have been made to determine whether dysbiosis was a cause or consequence of chronic inflammation. In IBD, for instance, patients display a dysbiotic profile [291,292,293]. Mice experiments revealed that in genetically identical hosts, the susceptibility to colitis relies on microbiota composition [294]. Recent studies have demonstrated that chronic inflammation also impacts microbiota at the meta-transcriptome level, in both IBD patients [295], and mouse colitis models [296,297]. Very recently, Becattini et al. assessed if gut microbiota also responds to acute (thus transient) host immune activation. In order to address this question, they used a simplified mouse model where GF mice were reconstituted with four anaerobic bacterial strains able to stably colonize the intestine [298]. Data indicates that following systemic immune activation via intraperitoneal flagellin administration (TLR5 activation), gut microbiota transcriptional and metabolic profiles rapidly change within 6 h, while the relative proportions of each strain remain stable. At 24h post immune activation, transcriptional profiles largely return to baseline. Flagellin administration triggers transcriptional modifications of hundreds of genes, upregulating genes involved in protein folding factors (chaperones), oxidative-species-scavenging molecules, and stress-response mediators, while downregulating genes mostly encoding metabolic enzymes, in particular, those involved in sugar catabolism and amino acid synthesis. Interestingly the use of another stimulus, e.g., anti-CD3 (T-cell activation), also triggers rapid and transient transcriptional modifications. Most of them are shared with flagellin conditions, but some differ, suggesting a gut microbiota adaption, depending on the stimulus and subsequent immune activation. Of note, when mice were reconstituted with three strains instead of four, the transcriptional profiles changes upon acute immune activation were different, suggesting that microbiota response to immunity also depends on its composition and on intra-microbes crosstalks. Altogether, those studies clearly indicate that immunity also deeply impact gut microbiota composition and function, depending on nature and lasting of the immune activation (i.e., acute versus chronic). The challenge is now to identify effector molecules involved in such dialogue, including immune system-derived metabolites.

### 3.2. The Antimicrobial Peptides

Antimicrobial peptides (AMPs) are key components of innate immunity against invading pathogens and represent one of the oldest innate effector systems. They act by disrupting bacterial cell membranes, modulating the immune response, and regulating inflammation. Being completely accurate, AMPs are not metabolites: They are small cationic molecules of 2–6 kDa that play an important role. The majority of intestinal AMPs is produced by specialized secretory cells of the small intestinal mucosa, namely, the Paneth cells. The expression of AMPs is tightly regulated by the presence of microorganisms via different mechanisms, mainly implicating the activation of PRRs (TLRs and NLRs) in IECs [299]. Of note, IL-22 further promotes their production [300]. Among the various AMPs produced in the GI tract, defensins and cathelicidins constitute the two major classes. It has been shown that GF mice display a reduced AMPs expression [301], and that Paneth cell alteration can lead to dysbiosis [302], underlining the importance of microbiota and Paneth cells for optimal AMPs production and subsequent homeostatic control of intestinal-associated microorganisms, respectively. It is noteworthy that some AMPs can be further metabolized into bioactive metabolites that can shape the microbiota. Ehmann et al. showed that the Human Defensin 5 can be further degraded by proteases into peptides displaying antimicrobial properties. Among them, one peptide fragment promoted *Akkermansia* sp. growth without decreasing microbial diversity in mice [303]. Thus, AMPs are molecules belonging to innate immunity that can profoundly shape microbiota. Further investigations are required to better characterize the in vivo impact of AMPs and derived bioactive products, to design new microbiota modulation strategies.

### 3.3. Immune Cells-Derived Metabolites and Metabolite Mimicry

Upon activation, effector immune cells display specific metabolic states (immunometabolism). Indeed, Glycolysis or OXPHOS are preferentially initiated by Teff- and Treg-cells, respectively [304]. Thus, immune cells both consume and produce metabolites that can presumably modify the micro-environment milieu, and thus, influence immune cells and bacterial functions. For instance, Treg-cells can consume extracellular ATP via CD39/CD73 ecto-ATPase activity, leading to Teff-cell dysfunction [305]. Another example, is the production of retinoic acid (RA), a 300 Da metabolite produced by epithelial cells and dendritic cells from retinol. Retinol and RA are derivatives of vitamin A, an essential fat-soluble micronutrient. Their production in the gut relies on their nutritional intake. RA, the bioactive form of vitamin A derivate, is known to modulate the immune system. Indeed, it is involved in T helper cell differentiation [306,307,308], it is a modulator of the innate immune cells [309,310], and a modulator of epithelial tight junction proteins [311,312]. The bioavailability of epithelium-derived RA depends on the microbiota. Intestinal bacteria regulate the expression of the Rdh7 gene of epithelial cells, coding for a retinol dehydrogenase which is involved in RA synthesis [313]. This example demonstrates that a metabolite produced by the intestinal epithelium following a stimulation of the microbiota can impact the immune response. Thus, the effect of RA on gut microbial communities relies on the indirect effect of RA regulating the immune system and mucosal barrier. It is thus conceivable that gut-associated immunity might trigger such metabolic-mediated effects on microbiota. Moreover, it has been shown that mammalian cells can produce molecules resembling bacterial metabolites. Ismail et al. described that colonic cells could produce an AutoInducer (AI)-2 mimic metabolite following tight-junctions disruption, or when submitted to a bacteria-derived secreted molecule [314]. AI-2 is proposed to promote interspecies bacterial communication [219], including in the mammalian gut. Interestingly, AI-2 mimic metabolite production by eukaryotic cells, and subsequent release within the gut lumen, modulates bacterial gene expression through AI-2 receptor signaling and activates quorum sensing system, including in pathogenic strains such as *Salmonella typhimurium* [314]. One can imagine that several metabolite mimics, able to modulate bacterial activity, might be produced by eukaryotic cells, including immune cells. It becomes now urgent to identify such immune-derived metabolites and metabolites mimics to better understand microbiota/host dialogue and design relevant strategies for manipulating these interactions.

## 4. Conclusions

This review provided a state-of-the-art description of the metabolites (produced from the microbiota, from the host, or from both) playing a major role in the crosstalk between gut microbiota and immune system.

Forty key metabolites are discussed, together with the modulations and immune pathways triggered in the gut. Evidence of the bidirectional impact of this crosstalk has been reviewed to contribute to a deeper understanding of the host-microorganism relationships, with a specific focus on the correlation between dysbiosis, microbial-based dysmetabolism, and pathogenesis.

To conclude, however extensive this review is, we should consider that 117 known microbial metabolites detected in fecal samples are present on the HMDB website. This suggests that we are at the dawn of the discovery of molecules that can have an impact on human health: Being part of this story is both exciting and challenging.

## 5. Challenges and Perspectives: How to Hear the Sound from Metabolite-Mediated Microbiota-Host Immunity Crosstalk

Research on metabolites in the gut has evolved extremely rapidly and enhanced our understandings of the crosstalk between host and microbiota in homeostasis and disease. Exploration of their richness and diversity requires special attention; in particular, the following five points should be carefully considered:Improve the analytical tools to detect metabolites in the gut (depth of data acquisition, detection of chemicals of various nature, dynamic range of detection with a large order of magnitude of detection). Currently, ultra-high performance liquid chromatography-high resolution mass spectrometry (UHPLC-HRMS) variants represent the best methods to address these challenges, and, in particular, to increase metabolite detection coverage [315]. The main limitations of LC-MS are the limited structural information and the complexity of obtaining absolute quantification, especially in non-targeted approaches.Correctly assign the identity of a metabolite. The vast majority of information collected by metabolomics is “dark matter”, i.e., chemical signatures that remain uncharacterized. Therefore, new computational solutions to illuminating dark matter are needed [316]. Significant progress has been reported in exploring public data-rich libraries, finding chemicals and associated metadata, and applying molecular networking strategies to accelerate metabolite annotation, especially in the last few years [317,318,319,320,321]. Moreover, the development and maintenance of MS-based spectral databases, together with the increasing practice of sharing metabolomics data through these resources, play a key role in translating dark matter into biological knowledge [322]. Authentic chemical standards should be used to acquire both positive and negative mode MS/MS and MS^n^ spectra to supplement spectral databases such as HMBD [323], NIST20 [324], METLIN [325], MZcloud [326], MassBank [327], ReSpect [328], and GNPS [329].Establish an extensive reference catalog of microbial genomes present in the human gut, as several big projects are already doing, such as the MetaHIT (METAgenomics of the Human Intestinal Tract) project; the Human Microbiome Project, or others [2,3].Integrate multi-omics datasets to recover microbe-metabolite relationships by using statistical analysis. Linear or neural network methods estimate the conditional probability that each molecule is present, given the presence of a specific microorganism [330,331,332]. The mmmvec tool (https://github.com/biocore/mmvec; accessed on 15 January 2021), for instance, can reliably identify all of the experimentally determined *P. aeruginosa*-produced molecules of interest in the lung of cystic fibrosis patient chronically infected with P. aeruginosa [331].Use experimental models relevant to the concept of molecular dialogue to unveil the respective role of microbiota and/or immune cells in the production of specific metabolites of interest. In a context where the microbial and host metabolite relationships are not elucidated, this first work consists of a detailed experimental design, followed by a fundamental study on metabolites using specific pathological models (KO mice, mono-colonization of the microbiota with certain strains, synthetic biology strategy, GF and specific pathogen-free model, wilding mice [333], culturomic, metabolomic model with minimal microbiota, (reviewed in [334]) and in vitro models [335,336].

Non-targeted metabolomics approaches can stratify the metabolome and to reveal notable differences from a “pathological” metabolome to a “physiological” metabolome. The detection and identification of metabotypes associated with the desired context, thus refining the reading of the metabolome and then encourage to associate its knowledge with previously observed cellular and molecular mechanisms or to set up experiments to define them.

The concept of exploiting metabolomics to perform activity screens to identify biologically active metabolites—which we term activity metabolomics—has been proposed recently by G. Siuzdak [337], and is already having a broad impact on biology. Finally, metabolomics investigation has shown to be a promising tool for the targeting and understanding of the microbiota-host crosstalk; however, this field is still in its infancy and more research and integrative tools are still needed.

## Figures and Tables

**Figure 1 metabolites-11-00406-f001:**
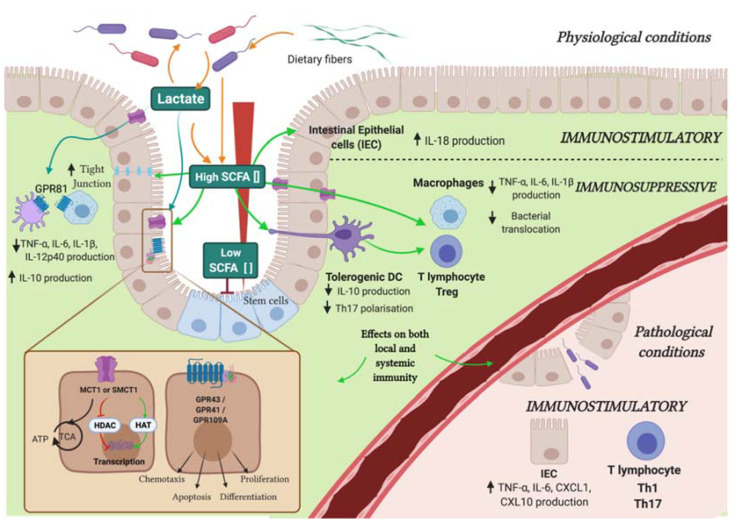
Role of short chain fatty acids (SCFAs) and lactate in immune regulation. SCFAs are produced from bacteria anaerobic fermentation of carbohydrates and dietary fibers. SCFAs have different modes of action on both local and systemic immunity. In physiological conditions, they regulate the intestinal barrier function by upregulation of the expression of the tight junction. Then, they play an important role in T-cell functioning, via regulation of G-protein-coupled receptors (GPR43, GPR41, GPR109A) and inhibition of HDAC (i.e., histone deacetylase). SCFAs regulate dendritic cells (DC) in the differentiation of T-cells, and Treg, Th1, and Th17 generation in different cytokine environments. Moreover, they inhibit the production of proinflammatory cytokines (e.g., TNF-α, IL-6, IL-1β) from intestinal macrophages and induce the production of interleukin-18 (IL18) from intestinal epithelial cells (IEC). In pathological conditions, SCFAs activate the production of proinflammatory cytokines and chemokines, such as Tumor Necrosis Factor α (TNF-α), Interleukin-6 (IL-6), CXCL1, and CXCL10, as well as increasing the production of Th1 and Th17 T lymphocytes. Lactate is a signal molecule of GPR81 and plays an important role in suppressing colonic inflammation.

**Figure 2 metabolites-11-00406-f002:**
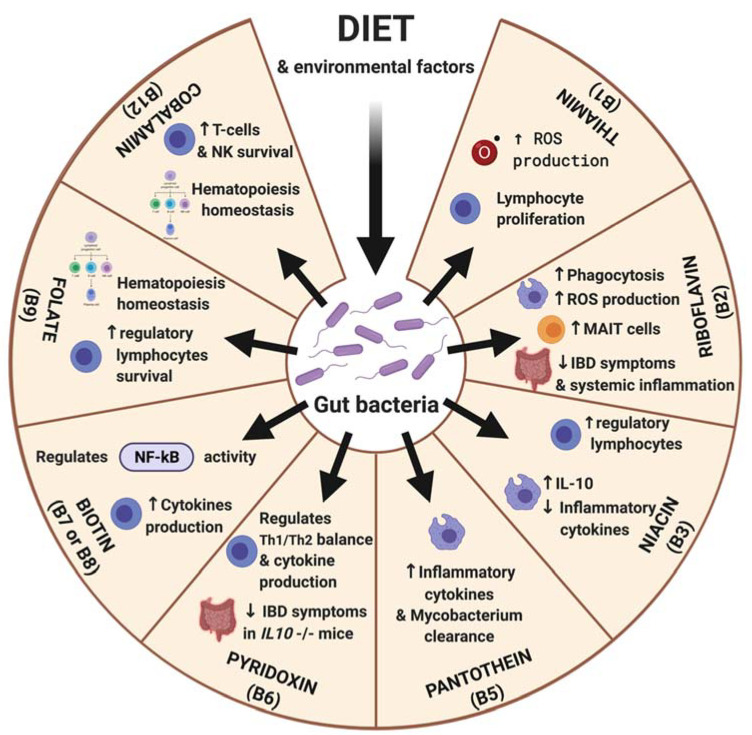
Overview of the impact of the different B vitamins on immunity. The main source of VB is the dietary intake, together with modification of dietary precursors or de novo synthesis carried out by the microbiota. As VB plays a role as precursors or as enzyme cofactors in numerous metabolic reactions, they impact the regulation of immune homeostasis through the metabolism, which is summarized here for each VB. MAIT, mucosa-associated invariant T lymphocytes; IBD, inflammatory bowel disease; ROS, reactive oxygen species).

**Figure 3 metabolites-11-00406-f003:**
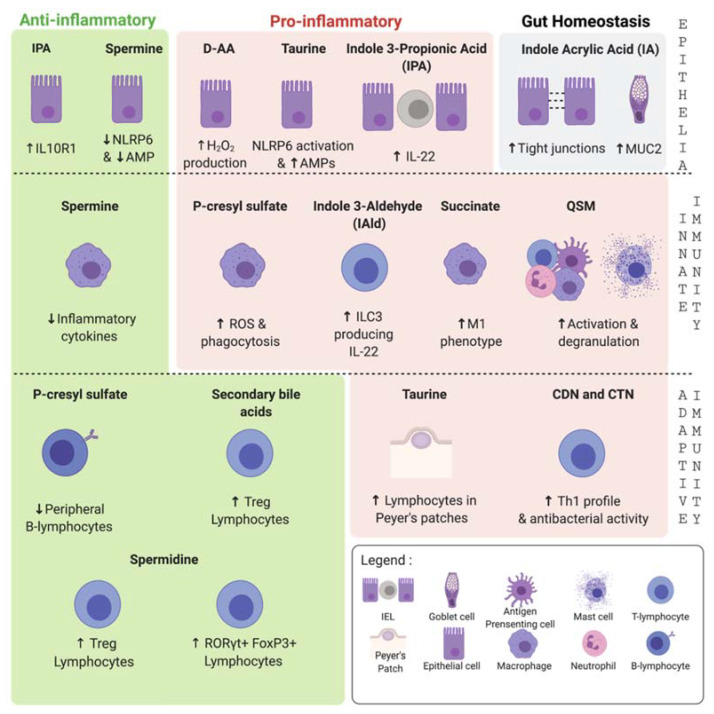
Summary of the immunomodulatory properties of the main described soluble metabolites on epithelial cells, as well as on innate and adaptive immune cells. D-AA, Stereoisomer D-amino acids; QSM, Quorum Sensing Molecules; CDN, Cyclic-DiNucleotides; CTN, Cyclic TriNucleotides.

**Figure 4 metabolites-11-00406-f004:**
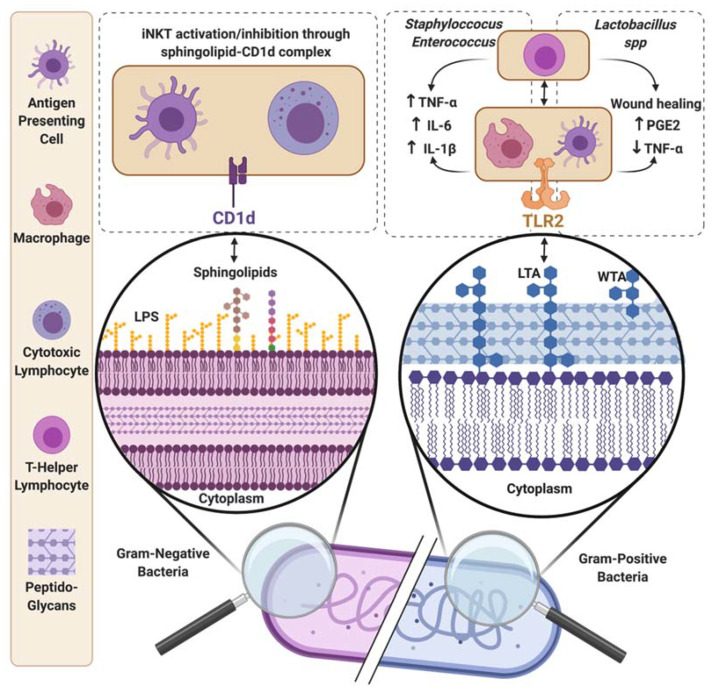
Summary of membrane-bound metabolites and their impact on immune cells. iNKT invariant Natural killer T, LTA, lipoteichoic acids; WTA, wall teichoic acids; PGE2, Prostaglandin E2.

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
