# Peer review of "What We Know So Far about the Metabolite-Mediated Microbiota-Intestinal Immunity Dialogue and How to Hear the Sound of This Crosstalk"

_metabolites, 2021, doi:10.3390/metabo11060406_

Round 1

Reviewer 1 Report

The authors present a very comprehensive and informative review on the metabolite-mediated microbiota-intestinal immunity dialogue and how to evaluate the contribution of microbial metabolites to the human metabolome. The authors discuss in detailed the contribution of key metabolites that known to have effects on immune responses and can contribute to a deeper insight into host-microorganism relationships. The manuscript also presents a literature survey for several important metabolites including substantial metabolites functions as well as metabolic pathways. Several metabolites classes including, Short-chain fatty acids, B Vitamins, Amino acids, and their derived metabolites, Catecholamines and Microbial membrane metabolites affecting the immune system are discussed in a very comprehensive and informative way. This review is very helpful for metabolomics community especially for graduate students and new researchers in the field of metabolomics. I would like to congratulate the authors for theirs great work and useful input in this timely review. I hope this review will encourage and help researchers to investigate this important and challenging topic in more deeply and inclusive studies.

I appreciate the fact that the author indicates in few sections the challenges and limitations of such kind of studies as stated in line 783 “This area of research is still emerging and we are certainly at the beginning of a concept

I consider this review as one the most important in the field and rank it as in top 5 percent. I recommend the paper for publication in its current status. How, over few comments and additions may improve the paper and make it more comprehensive

  • Add section on limitations and challenges and how to overcome these challenges in both experimental design and data processing.
  • Splitt the conclusion into two sections, one summary and future perspectives and a short conclusion.

Author Response

  • Split the conclusion into two sections, one summary and future perspectives and a short conclusion.

Per your suggestion, we split the conclusions in two sections: paragraph 4. Conclusions (including a brief summary of the review and a conclusive statement lines 1023 to 1035) and paragraph 5. Challenges and perspectives (including current challenges and future perspectives), beginning on line 1036.

  • Add section on limitations and challenges and how to overcome these challenges in both experimental design and data processing.

In the revised version of the manuscript, paragraph 5 includes real challenges in data processing and experimental design and suggestions on how to overcome these limitations, line 1096 to 1103 and 1050 to 1057 respectively with several references elaborating on these aspects that are not the focus of our review.

Reviewer 2 Report

The paper is novel and well written.

I have a few suggestions:

1) a general table summarizing metabolites of section 2 and 3 to get a clearer overview right away

2) the conclusions are too long and make the paper a little heavy 

Author Response

Response 1: We thank you for your time and appreciate that you have minor comments on the manuscript in its current form. You will find our answers point by point here:

  • a general table summarizing metabolites of section 2 and 3 to get a clearer overview right away

We have added a summarizing table and cite in the text line 123, as you suggested, and indeed we believe that this has improved the manuscript, as it enables a quick but detailed overview of the content of the review.

  • the conclusions are too long and make the paper a little heavy

As per your and the other reviewer’s suggestions, we split the conclusions in two sections: paragraph 4. Conclusions (including a brief summary of the review and a conclusive statement) and paragraph 5. Challenges and perspectives (including current challenges and future perspectives).
